# Solid Lipid Nanoparticles Loading Idebenone Ester with Pyroglutamic Acid: In Vitro Antioxidant Activity and In Vivo Topical Efficacy

**DOI:** 10.3390/nano9010043

**Published:** 2018-12-29

**Authors:** Lucia Montenegro, Anna Maria Panico, Ludovica Maria Santagati, Edy Angela Siciliano, Sebastiano Intagliata, Maria N. Modica

**Affiliations:** 1Department of Drug Sciences, University of Catania, 95125 Catania, Italy; panico@unict.it (A.M.P.); ludovica.santagati@icloud.com (L.M.S.); edysiciliano@hotmail.it (E.A.S.); s.intagliata@cop.ufl.edu (S.I.); 2Department of Medicinal Chemistry, College of Pharmacy, University of Florida, Gainesville, FL 32611, USA

**Keywords:** idebenone, solid lipid nanoparticles, nanocarriers, pyroglutamic acid, idebenone derivative, antioxidant activity, ORAC assay, anti-glycation activity, skin hydration

## Abstract

Idebenone (IDE), a strong antioxidant widely investigated for the treatment of neurodegenerative diseases and skin disorders, shows low oral and topical bioavailability due to its unfavorable physico-chemical properties. In this work, to improve IDE topical effectiveness, we explored a two-steps approach: (1) we synthesized an IDE ester (IDEPCA) with pyroglutamic acid, a molecule whose hydrating effects are well known; (2) we loaded IDEPCA into solid lipid nanocarriers (SLN). We evaluated in vitro antioxidant and anti-glycation activity and in vivo hydrating effects after topical application in human volunteers from gel vehicles of IDEPCA SLN in comparison to IDE SLN. All SLN showed good technological properties (mean particle size < 25 nm, polydispersity index < 0.300, good stability). The oxygen radical absorbance capacity assay showed that IDEPCA SLN and IDE SLN had similar antioxidant activity while IDEPCA SLN were more effective in the in vitro NO scavenging assay. Both IDEPCA and IDE SLN showed the same effectiveness in inhibiting the formation of advanced glycation end products. In vivo experiments pointed out a better hydrating effect of IDEPCA SLN in comparison to IDE SLN. These results suggest that the investigated approach could be a promising strategy to obtain topical formulations with increased hydrating effects.

## 1. Introduction

The advantages of solid lipid nanoparticles (SLN) as drug delivery systems have been widely reviewed [1,2,3,4]. High biocompatibility, the ability to load hydrophilic and lipophilic compounds, controlled release and drug targeting are some of the properties that have prompted researchers to explore the feasibility of using SLN to deliver several active compounds by different administration routes. In the last decade, a great deal of attention has been focused on SLN utilized as carriers for dermal delivery of antioxidants because of their ability to protect these molecules from degradation and to improve their topical bioavailability [5,6,7,8,9]. In a previous work, we reported a prolonged in vitro antioxidant activity upon loading idebenone (IDE) and IDE derivatives into SLN [10]. IDE is a synthetic analogue of CoQ_10_, which has been proposed both for the treatment of neurodegenerative diseases [11,12,13] and for topical use in the therapy of several skin disorders [14]. Owing to its unfavorable physico-chemical properties, IDE is regarded as a poor skin permeant that could benefit from loading into SLN to improve its topical effectiveness [15,16]. IDE loading into SLN proved to be able to target this antioxidant into the upper skin layers, avoiding its skin permeation in in vitro experiments [15]. In addition to favoring drug skin penetration, SLN have been reported to show an occlusive effect after topical application because of their ability to form an adhesive and continuous film on the skin surface that reduces transepidermal water loss, increasing skin hydration [17,18,19,20]. As reported in literature [21,22], maintenance of an optimal level of skin hydration depends on several factors among which the Natural Moisturing Factor (NMF), a heterogeneous blend of low molecular weight and water-soluble compounds formed within the corneocytes, plays a significant role. One of the main hydrating agents of NMF is pyroglutamic acid (2-pyrrolidone-5-carboxylic acid, PCA), a low molecular weight carboxylic acid showing extreme hygroscopic properties and hence, a great ability to bind water molecules. Recently, Jung et al. [23] pointed out an inverse relationship between PCA and caspase-14 amount in the stratum corneum and the concentration of inflammatory cytokines, whose expression is directly related to the severity of skin damages in atopic dermatitis patients [24,25,26]. Therefore, skin supplementation with PCA has been suggested as a valuable strategy in the treatment of skin disorders involving decreased skin hydration. The association between an antioxidant agent, which could reduce the inflammation process resulting from oxidative reactions, and PCA could provide a synergic effect in restoring the physiological skin functions. At present, the development of bifunctional molecules is regarded as a promising approach to improve the therapeutic outcome in comparison to a two-drug administration, reducing drug dosage and side effects [27,28]. Previously, to improve IDE effectiveness, we reported a two-step approach consisting of synthesizing ester derivatives by covalent linking IDE to other two antioxidants and subsequent loading of these molecules into SLN [10]. This strategy provided that IDE derivatives showed increased and prolonged antioxidant activity upon loading into SLN. Therefore, in this work, we adopted the same strategy by synthesizing IDE ester with PCA (IDEPCA) to increase the hydrating effect after topical application while maintaining the antioxidant activity. IDEPCA was loaded into SLN, whose good technological properties have been already reported [29,30]. The in vitro antioxidant activity of IDEPCA loaded SLN was compared with that of IDE loaded SLN and their hydrating effect was evaluated on human volunteers after topical application in gel formulations.

## 2. Materials and Methods

### 2.1. Materials

Glyceryl oleate (Tegin O^®^, GO), Cetyl palmitate (CP), Methylisothiazolinone and Methylchloroisothiazolinone (Acnibio AC^®^) and triethanolamine (TEA) were bought from ACEF (Fiorenzuola D’Arda, Italy). Idebenone (IDE) was obtained from Carbosynth (Berkshire, UK). Polyoxyethylene-20-oleyl ether (Brij 98^®^, Oleth-20), 2,2’-Azobis(2-methylpropionamidine) dihydrochloride (AAPH), fluorescin (FL) Trolox (TRL), aminoguanidine carbonate (AMG), bovine serum albumin (BSA), D-fructose, sodium nitroprusside and Griess reagent (1% dihydrochloride sulphanylamide and 0.1% naphthylethylenediamine (NED) dihydrochloride in 5% of hydrochloric acid) and sodium azide were purchased from Sigma-Aldrich srl (Milan, Italy). Curcumin was obtained from MP Biomedicals, (Illkirch cedex, France). Imidazolidinyl urea (Kemipur 100) was obtained from Farmalabor (Canosa di Puglia, Italy).

The following chemicals 2-Pyrrolidone-5-carboxylic acid (PCA), *N*,*N′*-dicyclohexylcarbodiimide (DCC), dry acetonitrile (CH_3_CN), 4-dimethylaminopyridine (DMAP), ethyl acetate (EtOAc), and hydrochloric acid (HCl), used to synthesize IDE ester (IDEPCA), were reagent grade and were bought from Sigma-Aldrich srl (Milan, Italy). DMSO-*d*_6_ used for NMR analysis was purchased from Sigma-Aldrich srl (Milan, Italy).

### 2.2. Chemistry

The synthesis of the ester IDEPCA was reported in Scheme 1. Commercially available Idebenone and PCA were reacted to obtain IDEPCA in the presence of DCC, a catalytic amount of 4-DMAP, and in dry acetonitrile as solvent.

Infrared spectrum was recorded on a Perkin Elmer series FTIR 1600 spectrometer (Milan, Italy). A sample droplet was placed between two disks of pure NaCl (neat sample). Signal intensity was characterized as s (strong), m (medium), w (weak). ^1^H NMR spectrum was performed on a Varian Inova Unity 200 spectrometer (200 MHz) (Milan, Italy) using DMSO-*d*_6_ as solvent. Chemical shifts are given in δ values (ppm), using tetramethylsilane as the internal standard; coupling constants (*J*) are given in Hz. Signal multiplicities are characterized as s (singlet), d (doublet), t (triplet), and m (multiplet). Reaction progress and purity of the synthesized compound were checked on TLC (an aluminum sheet coated with silica gel 60 F_254_, Merck, Darmstadt, Germany) using EtOAc as eluent and spots were visualized by UV light at 254 and 366 nm as the wavelength. IDEPCA was purified by flash column chromatography using Merck silica gel (0.040–0.063 mm).

### 2.3. Synthesis of (S)-10-(4,5-dimethoxy-2-methyl-3,6-dioxocyclohexa-1,4-dien-1-yl)decyl 5-oxopyrrolidine-2-carboxylate (IDEPCA)

A suspension of DCC (450 mg, 2.18 mmol) in dry CH_3_CN (10 mL) was added dropwise to a mixture of IDE (500 mg, 1.48 mmol), PCA (228 mg, 1.77 mmol), and DMAP (18 mg, 0.15 mmol) in dry CH_3_CN (25 mL), at 0 °C. Then, the mixture was stirred under nitrogen at room temperature for 3.5 h. The obtained precipitate of dicyclohexylurea was eliminated by filtration and the solvent was evaporated in vacuum to dryness. The residue was dissolved in EtOAc, washed with 0.5 N HCl (25 mL × 2), H_2_O (25 mL), and brine (25 mL). The organic phase was dried on anhydrous sodium sulphate, filtered, and evaporated in vacuum to dryness. The obtained oil was purified by flash column chromatography using EtOAc as eluent to give IDEPCA as a pure orange oil (270 mg, yield 40%).

IR (neat, selected lines) cm^−1^ 2927 (s), 2854 (m), 1742 (s), 1704 (s), 1651 (s), 1610 (s), 1456 (m), 1266 (s), 1204 (s), 1156 (w). ^1^H NMR (DMSO-*d*_6_) δ 1.20–1.36 (m, 14H, 7 CH_2_), 1.50–1.65 (m, 2H, CH_2_), 1.92–2.05 (s, 3H + 1H, CH_3_ + C*H*_A_H_B_), 2.11 (t, *J* = 4.2 Hz, 2H, CH_2_C=C), 2.22–2.42 (m, 2H + 1H, CH_2_C=O + CH_A_*H*_B_), 3.88 (s, 6H, OCH_3_), 4.07 (t, *J* = 6.6 Hz, 2H, CH_2_O), 4.16 (dd, *J* = 8.6 and 3.6, 1H, CH), 8.01 (s, 1H, NH). C_24_H_35_NO_7_.

### 2.4. Predicted Physico-Chemical Properties and Molecular Descriptors

A set of structural properties and molecular descriptors of IDE, PCA and IDEPCA were calculated using MarvinSketch, and reported in Table 1. The predicted physico-chemical properties include: molecular weight (MW), calculated partition coefficient (cLogP), hydrogen bond donor (HBD), hydrogen bond acceptor (HBA), topological polar surface area (TPSA), and rotatable bond number (RBN).

### 2.5. Determination of Water Solubility

IDE and IDEPCA water solubility was determined in triplicate by adding an excess of each compound to deionized water and stirring with a magnetic stirrer for 24 h at room temperature, while avoiding light exposure.

The resulting suspensions were filtered and the concentration of the compound in its saturated solution was determined spectrophotometrically (Shimadzu mod. UV-1601, Milan, Italy) at 278 nm for both IDE and IDEPCA. Prior to the solubility determination, a standard working curve was constructed from a known concentration of each compound in deionized water. The sensitivity of the assay was 0.5 μg/mL for IDE and IDEPCA.

Three aliquots of IDEPCA water saturated solution were stored at room temperature for 15 days to assess IDEPCA stability in aqueous medium. At intervals (0, 1, 3, 5, 7, 15 days), samples of these solutions were withdrawn and IDEPCA content was assayed spectrophotometrically as reported above.

### 2.6. Preparation of Solid Lipid Nanoparticles (SLN)

Solid lipid nanoparticles (SLN) were prepared by the phase inversion temperature (PIT) method, according to a procedure previously described [31]. For the preparation of these carriers, the following compounds were used in the oil phase: CP, oleth-20, GO, IDE and IDEPCA. The percentages (*w*/*w*) of each compound used to prepare such SLN are reported in Table 2. IDE was loaded into SLN at the maximum percentage used in topical formulations (1% *w*/*w*) and its derivative IDEPCA was used in an equimolar amount (1.33% *w*/*w*). The aqueous phase consisted of deionized water containing Acnibio AC^®^ 0.05% *w*/*w* and Kemipur 100 0.35% *w*/*w* as preservatives. Unloaded (SLN A) and IDE or IDEPCA loaded SLN were prepared using the same procedure. Briefly, the oil phase and the aqueous phase were separately heated and when both the oil and the water phase were at 90 °C, the aqueous phase was slowly added to the oil phase under stirring. Then, the mixture was allowed to cool to room temperature under continuous mixing and the phase inversion temperature (PIT) was recorded using a conductivity meter (model 525, Crison, Modena, Italy) when the turbid mixture turned into clear. Thin layer chromatography (TLC) analyses were performed to assess IDE and its derivative stability under these conditions, as previously reported [10]. TLC analyses were carried out as described in Section 2.2. Chemistry. TLC analyses pointed out that no degradation of IDE or IDEPCA occurred under these conditions.

SLN samples were stored in airtight jars at room temperature and in the dark until used.

### 2.7. Transmission Electron Microscopy (TEM)

For transmission electron microscopy (TEM) analysis, 5 μL of SLN samples were placed on a Formvar (200-mesh) copper grid (TAAB Laboratories Equipment, Berks, UK). When all of the sample was absorbed, the excess was removed by filter paper and a drop of 2% (*w*/*w*) aqueous solution of uranyl acetate was added over 2 min. Before starting the analysis with a transmission electron microscope (model JEM 2010, Jeol, Peabody, MA, USA) operating at an acceleration voltage of 200 KV, the surplus was removed and the sample was allowed to dry at room temperature.

### 2.8. Photon Correlation Spectroscopy (PCS)

The mean particle size and the size distribution (polydispersity index, PDI) were determined by dynamic light scattering (DLS) using a Zetasizer Nano ZS90 (Malvern Instruments, Malvern, UK), using a 4 mW laser diode at 670 nm and scattering light at 90°.

All SLN were analyzed after dilution (1:5, sample/distilled water) and sample were adapted to 25 °C for 2 min prior to the analysis. The measurement were carried out in triplicate and the results were expressed as mean ± SD. ζ-potential was determined by laser Doppler velocimetry using the same Zetasizer reported above, after diluting each sample with KCl 1 mM (pH 7.0), according to a procedure previously reported [32].

### 2.9. Stability Tests

Particle sizes, PDI and ζ potential values of SLN samples were measured at intervals (24 h, one week, two weeks, one month, two months). During storage, samples were maintained at room temperature and protected from light exposure.

### 2.10. Oxygen Radical Absorbance Capacity (ORAC) Assay

The antioxidant activity of IDE and IDEPCA, free or loaded into SLN, was assessed in vitro using the oxygen radical absorbance capacity (ORAC) assay, according to the procedure previously reported by Cao et al. [33,34]. Data were obtained using a VICTOR Wallac 1420 Multilabel Counters fluorimeter (Perkin Elmer, Boston, MA, USA) with a fluorescence filter (excitation 540 nm, emission 570 nm). Fluorescein (FL) solution (12 nM) was the fluorescence probe and the target molecule for free radical attack by the peroxyl radical generator AAPH (100 mM). The assay was carried out at pH 7.0 and at 37 °C, using trolox (12.5 μM) as the control standard and phosphate buffer (pH 7.0) as the blank. IDE and IDEPCA were solubilized in ethanol (12.5 µM) while IDE or IDEPCA loaded SLN were diluted (12.5 µM) with phosphate buffer. After adding AAPH, the fluorescence was recorded every 2 min. All measurements were expressed in relation to the initial reading, analyzing all samples, one blank and one standard at the same time. Each measure was performed in triplicate. The ORAC value refers to the net protection area under the quenching curve of FL in the presence of an antioxidant. The results (ORAC values) were calculated and were expressed using trolox equivalents (TE) for μM of sample (TE/μM) according to Equation (1):ORAC units (TE/μM) = K (*S*_sample_ − *S*_blank_)/(*S*_trolox_ − *S*_blank_)(1)
where K is a sample dilution factor and *S* is the area under the fluorescence decay curve of the sample, trolox, or blank calculated with Origin^®^7 (OriginLab Corporation, Northampton, MA, USA).

### 2.11. NO Scavenger Assay

The ability of IDE or IDEPCA, free or loaded into SLN (1 mg/mL), to inhibit the spontaneous NO production from an aqueous solution of sodium nitroprusside (20 mM) at 25 °C for 3 h was evaluated using Griess reagent [35]. Curcumin (100 µg/mL) was used as reference compound [36]. Absorbance was measured at 540 nm with a spectrophotometer (Thermo Scientific MultiskanR EX.). The percent of inhibition of NO radical production was calculated according to the following equation:% of inhibition of NO = [*A*_0_ − *A*_1_]/*A*_0_ × 100(2)
were *A*_0_ is the absorbance (nm) of untreated sample, *A*_1_ (nm) is the absorbance (nm) of treated samples (IDE, IDEPCA, SLN IDE, SLN IDEPCA, SLN A).

### 2.12. Anti-Glycation Activity

Anti-glycation ability was evaluated determining the inhibition of fluorescence produced by advanced glycation end products (AGEs) formation through Maillard reaction, using the method of Derbre et al. [37]. Protein model bovine serum albumin (BSA) (10 mg/mL) was incubated with D-fructose (0.5 M) in phosphate buffer 50 mM, pH 7.4 and NaN_3_ 0.02% *w/v* as positive controls. BSA alone was used as negative control as it did not provide any formation of fluorescent AGEs. Aminoguanidine (AMG) (3 mM) was used as the reference compound [38]. Final glycated BSA solution (300 μL) alone and with IDE or IDEPCA, free or loaded into SLN (2 mg/mL) was incubated at 37 °C in 96-well microtiter closed with their silicon lids for 7 days. Inhibition of fluorescence was determined using a VICTOR Wallac 1420 Multilabel Counters fluorimeter (Perkin Elmer, USA) (λexc 370 nm; λem 440 nm). Results were reported as relative fluorescence units (RFU) and the percentage of inhibition with respect to the positive control (BSA with fructose) were calculated from the following equation:% of inhibition = {1 − [RFU sample (nm)/RFU Positive control (nm)]} × 100(3)

### 2.13. Gel Preparation

The composition of gel formulations is reported in Table 3. Carbopol Ultrez 21 (0.8% *w*/*w*) was used as rheological additive and TEA (1.00% *w*/*w*) for neutralization. Carbopol Ultrez 21 was dispersed in the aqueous phase consisted of deionized water containing 0.35% *w*/*w* Kemipur 100 and 0.05% Acnibio AC (gel C), unloaded SLN (gel SLN), 0.32% *w*/*w* PCA in deionized water (gel PCA), IDE loaded SLN (gel IDE), IDEPCA loaded SLN (gel IDEPCA), IDE loaded SLN and 0.32% *w*/*w* PCA (gel IDE/PCA). Owing to IDE poor water solubility, we could not prepare a gel containing only free IDE as control, even when adding low amounts of solubilizing agents to the vehicle. Full polymer hydration was obtained by maintaining the mixture in the dark at room temperature for 24 h. Formulations were made viscous by adding TEA (1.00% *w*/*w*) drop by drop, under slow mixing to avoid air bubbles formation. All gels were stored in airtight jars at room temperature and in the dark to protect from photo-degradation until use.

### 2.14. In Vivo Evaluation of Gel Formulations

To determine the in vivo efficacy of SLN containing IDE or IDEPCA from gel formulations, twelve female subjects with average photo aging were selected (age range 40–60 years). Investigations were performed according to the rules of the Declaration of Helsinki of 1975, revised in 2008. The local ethics committee declared that no approval was needed for this type of study, due to its nature and the safety of the formulations under investigation.

After receiving the required information about the nature of the study, all volunteers provided their written informed consent. Four different gels, randomly chosen among the six prepared gels, were supplied to each volunteer in containers labeled with color codes (double blind study).

The volunteers were instructed to apply each gel (about 2 mg/cm^2^) into different areas over the back of the hands, twice a day, morning and evening, for one week, without applying any other product.

Measurements of skin hydration have been determined before the application of the products (baseline value) and after one week of treatment, using the specific probes of the instrument Soft Plus (Callegari Srl, Parma, Italy). Using these probes, skin hydration was assessed by capacity measurements in the range 0–100 u.c. (arbitrary units) with a resolution of 1 u.c. and a 5% precision. All data were acquired under controlled temperature (22 ± 1 °C) and humidity (35% ± 5%) conditions. Each measurement was performed in triplicate. Skin hydration values were reported as difference between the values obtained before and after applying the samples under investigation (Δ hydration). Data were expressed as mean ± S.D. Statistical analysis of the results was performed using Students’ test (*p* < 0.05).

## 3. Results and Discussion

### 3.1. Solid Lipid Nanoparticle Characterization

Unloaded and IDE or IDEPCA loaded SLN, prepared by the PIT method, showed small particle sizes (about 24 nm), which were not affected by IDE or IDEPCA loading (see Table 4). PDI values were lower than 0.300, suggesting the presence of a homogeneous population of nanoparticles in the investigated samples.

Morphological analyses pointed out the presence of spherical particles whose sizes roughly matched those determined by DLS (Figure 1). All these colloidal systems proved stable after storage for two months at room temperature (data not shown) despite their slightly negative ζ potential values (−1.83/−2.02 mV). Similar low ζ potential values were previously obtained for SLN prepared by the PIT method, which showed long-term stability [29,30,31].

The PIT or HLB (hydrophilic lipophilic balance) temperature has been reported to be an indicator of the stability of colloidal systems as the higher the PIT value, the higher the formulation stability (Izquierdo et al. [39]). As shown in Table 4, the PIT values of loaded SLN were greater than the ones observed for unloaded SLN while their ζ potential values were not significantly different. A similar behavior has been previously reported for SLN loading IDE derivatives [10,15,31], thus confirming that, apart from ζ potential, the PIT values were a key factor for the stability of SLN.

Loaded SLN were prepared using equimolar amounts of IDE and IDEPCA. According to the data reported in Table 1, cLogP value of IDE was greater than that of IDEPCA (2.25 and 1.94, respectively); nevertheless, the water solubility of IDE resulted greater than that of IDEPCA (15.7 µg/mL vs. 11.7 µg/mL, respectively). However, further structural properties (i.e., number of HBD, HBA, and MW), as well as the intrinsic properties of each functional group of the molecule, could also affect the overall water solubility. In general, water solubility tends to decrease with increasing MW, thus the higher MW of IDEPCA could reduce its solubility in water. In addition, the higher the number of total hydrogen bonds present in the molecule, the greater the chance that the compound would be water soluble, even though the formation of intramolecular hydrogen bonds might mitigate this effect. Specifically, the hydroxyl group of IDE is able to participate in intermolecular hydrogen bonding with water molecules, while the ester group of IDEPCA can form intramolecular hydrogen bonds with the amide group of the PCA moiety instead, decreasing the water solubility. The molecular descriptors reported in Table 1 showed that IDE, PCA, and IDEPCA possess MW less than 500 daltons, cLogP and HBD less than 5, HBA lower than 10. These descriptor values for all compounds satisfy the Lipinski’s rule of five. In addition, the low molecular flexibility (TPSA < 140 Å) of all compounds contributes to the respect of the Lipinski’s rule in predicting a good oral bioavailability in rat, even if the RBN value is more than seven for IDE and IDEPCA. Therefore, IDEPCA could be a good candidate as an orally active drug. On the other hand, Ates et al. [40] reported that high dermal absorption could be predicted when at least two of the following parameters are triggered: MW < 180 Da, log P ≥ 0.3, MP < 100 °C, TPSA < 40 Å. As reported in the chemical data sheet provided by the manufacturer, IDE melting point is 53–55 °C while IDEPCA is an oil at room temperature. Therefore, according to the data illustrated in Table 1, both IDE and IDEPCA triggered two of the above-mentioned parameters (log P ≥ 0.3, MP < 100 °C) and could be regarded as high skin permeants. In addition, as reported in literature [41], the lower the melting point, the higher the skin permeation ability. Therefore, IDEPCA could be predicted as a better skin permeant than IDE owing to its lower melting point.

IDEPCA proved stable in water during storage at room temperature for 15 days as no significant change of IDEPCA content was observed during this period (data not shown).

### 3.2. In Vitro Antioxidant and Anti-Glycation Activity

Scientific evidence shows that glycation along with oxidative processes are major responsible for both several diseases and related risk of complications, especially cardiovascular, and aging processes [42]. Glycation leads to the formation of advanced glycation end products (AGEs), whose accumulation is associated with endothelial dysfunctions and inflammatory processes. Collagen binding and formation of highly crosslinked structures as well as increase in inflammatory mediators and oxygen and nitrogen reactive substances are regarded as AGEs mechanisms resulting in the impairment of body physiological functions. In addition, AGEs bind to specific receptors (RAGE), inducing oxidative stress and promoting inflammation. Hence, functional and structural changes of glycan are regulated from the generation of reactive oxygen metabolites (ROMs) such as reactive oxygen species (ROS) or reactive nitrogen species (RNS). ROS and RNS produced in the mitochondria cause damage to macromolecules including lipids, proteins and mitochondrial DNA [43,44]. The accumulation of this damage contributes to the development of various diseases, especially diseases associated with aging. Experimental studies suggest that oxidative modifications of the endothelium are involved in functional and structural changes leading to endothelial dysfunctions [45]. Therefore, the preservation of the endothelial antioxidant capacity and the inhibition of peroxidation processes are regarded as attractive strategies to prevent related diseases and ageing. Antioxidants are useful to reduce and to prevent damage caused by free radical reactions [46] owing to their ability to regulate the transfer of electrons or to quench free radicals escaping from the electron transport chain, thus reducing the number of free radicals and consequently the damage responsible for ageing and degenerative diseases. Several in vitro and in vivo assays have been proposed to estimate the antioxidant capacity of natural and synthetic compounds [47,48]. In this study, we assayed the antioxidant effect of IDE and its derivative IDEPCA, both in free form and loaded into SLN, by evaluating their oxygen-radical absorbance capacity (ORAC) and nitric oxide (NO) scavenger activity. In addition, we explored their anti-glycant effect on AGEs formation.

The ORAC assay has emerged as a valuable analytical method to determine the antioxidant potential of a range of substances [49,50], representing a reaction mechanism relevant to human biology [51]. This assay evaluates the fluorescence decrease of a fluorescent probe (fluorescein) due to the action of peroxyl radicals generated by thermal decomposition of 2,2′-azobis(2-methylpropionamidine) dihydrochloride (AAPH) [33]. In the presence of an antioxidant, peroxyl radicals are scavenged and the decay of the fluorescence curve is retarded. In our experiments, the difference between the “area under the fluorescence decay curve” (AUC) in the presence and in the absence of an antioxidant was translated into a Trolox standard calibration curve to express the antioxidant activity as Trolox equivalent for µM (TE/µM) of samples.

As illustrated in Table 5, free IDE and its ester derivative, IDEPCA, showed an antioxidant activity greater than that of the reference standard Trolox. In particular, IDEPCA antioxidant activity was greater than that of IDE, suggesting that the derivatization of the IDE with PCA may result in changes in the electronic arrangement of the molecule, leading to a different ability to interact with the peroxyl radical that was produced during the ORAC assay. PCA, used to obtain the ester IDEPCA, did not exhibit any antioxidant activity against the peroxyl radical. Unloaded SLN showed an ORAC value similar to Trolox. Analogous results have been already reported in a previous paper [10] evaluating the antioxidant activity by ORAC assay of unloaded SLN with the same composition. The antioxidant activity of these unloaded SLN was attributed to the presence of oleic residues in SLN components (oleth-20 and glyceryl oleate) [52]. Loading IDE or IDEPCA into SLN provided only a slight but statistically significant increase of antioxidant activity in comparison to empty SLN. As shown in Table 5, although free IDEPCA showed greater antioxidant activity than free IDE, loading these compounds into SLN provided similar ORAC values. This behavior could be due to a slow release of the active molecules (IDE and IDEPCA) from the SLN matrix, resulting in low amounts of free molecules available to act as antioxidant. As no ORAC value could be detected for PCA, SLN loading IDE and PCA, in the same molar ratio of IDEPCA, were not assessed as no increase of antioxidant activity could be expected in comparison to SLN loading only IDE.

Nitric oxide (NO) is an important chemical mediator involved in the regulation of various physiological processes whose concentration increase is associated with several diseases [53]. In this study, we determined the NO scavenging capacity of IDE and IDEPCA, free or loaded into SLN, evaluating their ability to inhibit the spontaneous in vitro NO production from an aqueous solution of sodium nitroprusside, as a NO donor, using Griess reagent [35]. As shown in Table 5, the results of this assay were expressed as a percentage of NO scavenging. The ability of IDE or IDEPCA, as free form or loaded into SLN, to reduce NO production was lower than Trolox. IDE or IDEPCA loaded SLN showed a NO scavenging ability lower than the corresponding free active ingredient, likely because of a slow release of these antioxidants from the lipid matrix of the SLN.

AGEs are a group of highly complex oxidant compounds produced by metabolic reactions including the Maillard reaction, generated in vivo as response to an increased oxidative stress. The results of the in vitro anti-glycation assay pointed out that both IDE and IDEPCA showed a capacity to inhibit AGEs formation significantly higher than Trolox and their incorporation into SLN provided a two-fold increase of this activity, with percentages of inhibition higher than 90%. The lower ability to inhibit AGEs formation of free IDE in comparison to IDE loaded into SLN could be due to the high protein binding of this antioxidant [54], resulting in low amounts of free IDE available to inhibit the glycation process. Loading IDE into SLN prevented it from being bound to proteins and allowed its slow release in the medium. As the glycation process went on, the fraction of unmodified albumin available to IDE binding decreased. Therefore, during the long incubation time of this experimental procedure (7 days), the amount of IDE (or IDEPCA) released from the carriers could have outweighed the antioxidant binding to albumin, thus resulting in the increased AGEs inhibition observed for IDE or IDEPCA loaded SLN. To verify this hypothesis, further studies have been planned to elucidate the mechanism of inhibition of AGEs formation due to IDE and IDEPCA loaded SLN.

### 3.3. In Vivo Evaluation of Gel Formulations

Gels and emulsions are the most frequently used vehicles for topical application of lipid nanoparticles [55]. Souto et al. [56] highlighted that SLN and NLC particles size and zeta potential were not affected by incorporation in hydrogels. Therefore, to avoid possible interactions between vehicle components and lipid nanocarriers, which may occur in complex vehicles such as emulsions, we incorporated unloaded and IDE, IDEPCA and IDE/PCA loaded SLN into gel formulations. We loaded 1% *w*/*w* IDE into SLN as this is the highest concentration used in marketed products and has already been tested in lipid nanocarriers for topical delivery of IDE [57]. The values of skin hydration (Δ hydration), obtained after one-week in vivo topical application with the gels under investigation, are shown in Figure 2. Gel C, which did not contain PCA or lipid nanocarriers, was used as control for in vivo evaluation and did not provide an increase of skin hydration. Free IDE or IDEPCA were not used as control due to their poor water solubility, which prevented us from incorporating them into an aqueous gel without the addition of co-solvents that could affect skin permeability.

Topical application of gel PCA and gel SLN (containing unloaded SLN) resulted in an improvement of skin hydration and no significant difference was observed between these two formulations. As shown in Figure 2, IDE loading into SLN did not lead to an increase of Δ hydration in comparison to unloaded SLN while IDEPCA loaded SLN remarkably improved this parameter and a significant difference (*p* < 0.05) was observed with respect to all other gels. To evaluate the influence of IDE esterification with PCA, a gel containing a physical mixture of IDE loaded SLN and PCA (in the same molar ratio of IDEPCA) was assessed for its ability to improve skin hydration. This formulation provided an effect lower than IDEPCA loaded SLN but similar to that observed for gels incorporating unloaded or IDE loaded SLN or containing only PCA.

The highest hydrating ability of gel IDEPCA could be attributed to a better capacity of IDEPCA to penetrate into the skin layers in comparison to IDE. The better hydrating effect of IDEPCA loaded SLN is supported by previous patents on PCA esters designed to improve skin hydration [58,59]. Further studies have been planned to evaluate IDEPCA loaded SLN effects on other cutaneous parameters, such as skin elasticity, and to investigate their interactions with the skin tissue.

## 4. Conclusions

IDE esterification with pyroglutamic acid provided an IDE derivative (IDEPCA) with improved in vitro antioxidant activity in comparison to IDE. IDE or IDEPCA loading into SLN resulted in a decrease of antioxidant activity while the ability to inhibit AGEs formation notably increased. In vivo studies performed on human volunteers utilizing IDE or IDEPCA SLN incorporated in gel vehicles pointed out a greater hydrating effect of IDEPCA SLN in comparison to IDE SLN. These results suggest that the synthesis of ester derivatives with pyroglutamic acid and their loading into SLN could be a valuable approach to develop topical formulations with increased hydrating effects.

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
