# Peer review of "Solid Lipid Nanoparticles Loading Idebenone Ester with Pyroglutamic Acid: In Vitro Antioxidant Activity and In Vivo Topical Efficacy"

_nanomaterials, 2018, doi:10.3390/nano9010043_

Round 1
Reviewer 1 Report
The reported work aims at achieving synergic effects in restoring the skin functions by encapsulation of bifunctional molecules in solid lipid nanoparticles.
The manuscript needs the following revisions before publication in the journal:
1) Please use the full name of IDEPCA in the Abstract.
2) The Introduction should focus also on the composition and structures of the nanocarriers and not only on the properties of the bioactive molecules.
It should mention references on other types of nanocarriers for delivery of insoluble compounds, for instance liquid crystalline nanoparticles encapsulating insoluble drugs:
- Exp Dermatol. 2015, 24, 449-454. DOI: 10.1111/exd.12696 .
- Advances in Colloid and Interface Science, 2017, 249, 331-345. DOI: 10.1016/j.cis.2017.04.006 ;
- J. Mater. Chem. B, 2015, 3, 7734-7744, DOI: 10.1039/C5TB01193K ;
- Neural Regeneration Research, 2017, 12, 886-889 , DOI:10.4103/1673-5374.208546 ;
- Pharm Res. 2017, 34, 492–505, DOI: 10.1007/s11095-016-2080-4 ;
- Soft Matter, 2016, 12, 7539-7550. DOI: 10.1039/c6sm00661b ;
- Eur J Pharm Biopharm. 2014, 86, 121-132. DOI: 10.1016/j.ejpb.2013.12.011.
3) The legend of Scheme 1 is not complete: The main molecules are not indicated.
DCC, DMAP, and CH3CN are not visible in the Scheme.
4) Table 2 can indicate the entire composition of the formulations.
What is the wt% of the water phase with regard to the wt% of the oil phase in the formulations?
Brij98 is a more common name, which may replace Oleth-20 in the Table.
5) Methods: The rationale of studying 1% drug loading should be better justified. What is the maximal percentage of drug loading in the SLN?
6) The TEM image of IDEPCA-loaded SLNs is missing in Figure 1.
7) page 9, lines 354-375: The text of #3.2 sounds as a literature research report rather than as a presentation of new results. Please include in #3.2 first the Figures showing the results.
8) page 10, line 386: The figure with the Fluorescence curves is missing and must be included.
9) English
- p.3, line 118: The sentence must begin with a subject followed by a verb.
- p.5, line 163: compounds
- p.5, line 189: Photon Correlation Spectroscopy
Author Response
Reviewer 1
The reported work aims at achieving synergic effects in restoring the skin functions by encapsulation of bifunctional molecules in solid lipid nanoparticles.
The manuscript needs the following revisions before publication in the journal:
1) Please use the full name of IDEPCA in the Abstract.
Answer
According to the instructions for authors, abbreviations should be defined in parentheses the first time they appear in the abstract, main text, and in figure or table captions and used consistently thereafter.
We reported the full name of the ester derivative investigated in this work (idebenone ester with pyroglutamic acid) in line 16 and we defined in parentheses the meaning of the abbreviation IDEPCA the first time it appeared in the abstract (line 16). We reported the chemical (IUPAC) name of IDEPCA in section 2.3. “synthesis”. It is unclear what the reviewer meant asking us to use the full name of IDEPCA in the Abstract.
2) The Introduction should focus also on the composition and structures of the nanocarriers and not only on the properties of the bioactive molecules.
It should mention references on other types of nanocarriers for delivery of insoluble compounds, for instance liquid crystalline nanoparticles encapsulating insoluble drugs:
- Exp Dermatol. 2015, 24, 449-454. DOI: 10.1111/exd.12696 .
- Advances in Colloid and Interface Science, 2017, 249, 331-345. DOI: 10.1016/j.cis.2017.04.006 ;
- J. Mater. Chem. B, 2015, 3, 7734-7744, DOI: 10.1039/C5TB01193K ;
- Neural Regeneration Research, 2017, 12, 886-889 , DOI:10.4103/1673-5374.208546 ;
- Pharm Res. 2017, 34, 492–505, DOI: 10.1007/s11095-016-2080-4 ;
- Soft Matter, 2016, 12, 7539-7550. DOI: 10.1039/c6sm00661b ;
- Eur J Pharm Biopharm. 2014, 86, 121-132. DOI: 10.1016/j.ejpb.2013.12.011.
Answer
The reviewer asked us to focus our attention on liquid crystalline nanoparticles and to add several references. The reviewer reported only journal and doi of these references without specifying authors and titles.
We have found all these references and we have listed them below.
Esposito E, Sticozzi C, Ravani L, Drechsler M, Muresan XM, Cervellati F, Cortesi R, Valacchi G. Effect of new curcumin-containing nanostructured lipid dispersions on human keratinocytes proliferative responses. Exp Dermatol. 2015, 24, 449-54. doi: 10.1111/exd.12696.
Angelova A, Garamus VM, Angelov B, Tian Z, Li Y, Zou A. Advances in structural design of lipid-based nanoparticle carriers for delivery of macromolecular drugs, phytochemicals and anti-tumor agents. Adv Colloid Interface Sci. 2017, 249, 331-345. doi: 10.1016/j.cis.2017.04.006.
Chen Y, Angelova A, Angelov B, Drechsler M, Garamus VM, Willumeit-Römere R, Zou A. Sterically stabilized spongosomes for multidrug delivery of anticancer nanomedicines. J. Mater. Chem. B, 2015, 3, 7734-7744. doi: 10.1039/C5TB01193K.
Angelova A, Angelov B. Dual and multi-drug delivery nanoparticles towards neuronal survival and synaptic repair. Neural Regen Res. 2017, 12,, 886-889. doi: 10.4103/1673-5374.208546.
Guerzoni LP, Nicolas V, Angelova A. In Vitro Modulation of TrkB Receptor Signaling upon Sequential Delivery of Curcumin-DHA Loaded Carriers Towards Promoting Neuronal Survival. Pharm Res. 2017, 34, 492-505. doi: 10.1007/s11095-016-2080-4.
Zerkoune L, Lesieur S, Putaux JL, Choisnard L, Gèze A, Wouessidje D, Angelov B, Vebert-Nardin C, Doutchh J, Angelova A. Mesoporous self-assembled nanoparticles of biotransesterified cyclodextrins and nonlamellar lipids as carriers of water-insoluble substances. Soft Matter, 2016, 12, 7539-7550. doi: 10.1039/c6sm00661b
Esposito E, Ravani L, Mariani P, Huang N, Boldrini P, Drechsler M, Valacchi G, Cortesi R, Puglia C. Effect of nanostructured lipid vehicles on percutaneous absorption of curcumin. Eur J Pharm Biopharm. 2014 Feb;86(2):121-32. doi: 10.1016/j.ejpb.2013.12.011.
Looking at these references, it is evident that the same author appears in five papers out of seven. Another research group published the remaining two manuscripts.
It is unclear why we should focus our attention on liquid crystalline nanoparticles and we were advised to add these references. At present, many drug delivery systems (liposomes, polymeric nanoparticles, micro and nanoemulsions, dendrimers, cyclodextrins etc.) that could be used as carriers for insoluble drugs are available and thousands of articles have been published dealing with these nanocarriers. In addition, the relevance to the topic of our work of most of the above-mentioned references is questionable. It is unclear how the addition of these references could improve the quality of our manuscript or could provide the reader with better information on the topic of the present manuscript.
Therefore, we have not added the above-mentioned references and we have not included liquid crystalline nanoparticles in the text of our manuscript.
3) The legend of Scheme 1 is not complete: The main molecules are not indicated.
Answer
To comply with the reviewer’s request, we added the main molecules in the legend of Scheme 1.
DCC, DMAP, and CH3CN are not visible in the Scheme.
Answer
DCC, DMAP, and CH3CN have been reported in the legend of Scheme 1. We added the letter a) in Scheme 1 to make clear that reagents and conditions are reported in the legend of Scheme 1. The legend of Scheme 1 has been modified accordingly.
4) Table 2 can indicate the entire composition of the formulations.
What is the wt% of the water phase with regard to the wt% of the oil phase in the formulations?
Answer
The amount of water used to prepare each formulation has been reported as footnote: water phase q.s. 100, which means that we added to the oil phase the amount of water necessary up to 100 %. We added in the footnote the notation “%” to make this concept clearer.
Brij98 is a more common name, which may replace Oleth-20 in the Table.
Answer
Brij98 is a commercial name while oleth-20 is the INCI (international Nomenclature of Cosmetic Ingredients) name of polyoxyethylene-20-oleyl ether. As this molecule is marketed under different brand names, we prefer to use its INCI name.
5) Methods: The rationale of studying 1% drug loading should be better justified. What is the maximal percentage of drug loading in the SLN?
Answer
As suggested by the reviewer, we have added the following comment in the text (line 449) to explain better the use of idebenone 1% w/w:
We loaded 1% w/w IDE into SLN as this is the highest concentration used in marketed products and has already been tested in lipid nanocarriers for topical delivery of IDE [57].
Idebenone and IDEPCA loading into the investigated SLN was 1.5% and 2%, respectively. However, as we believe that the use of concentrations greater than those used in marketed products could be questionable, we have not reported this data in the manuscript.
6) The TEM image of IDEPCA-loaded SLNs is missing in Figure 1.
Answer
To comply with the reviewer’s request, the TEM image of IDEPCA loaded SLN has been added in Fig. 1.We have modified the legend of Fig. 1 accordingly.
7) page 9, lines 354-375: The text of #3.2 sounds as a literature research report rather than as a presentation of new results. Please include in #3.2 first the Figures showing the results.
Answer
Section 3.2. is part of the Results and Discussion section of the manuscript. Before presenting and discussing our results, we have briefly introduced the main concepts involved in the oxidation and glycation processes as the reader is not supposed to be an expert in this field. We could deal with these fundamental aspects of these processes in the introduction. However, we have preferred to insert these concepts in the Results and Discussion section to avoid an excessively long introduction. Therefore, we have not changed the text of section 3.2.
8) page 10, line 386: The figure with the Fluorescence curves is missing and must be included.
Answer
In the manuscript, we reported the ORAC values of all investigated samples. The ORAC assay is a well-known test that is routinely used to determine the antioxidant activity. The addition of the fluorescence curves in the text would not provide the reader with additional information as the ORAC values cannot be easily extrapolated from these curves. Therefore, we have not added these curves in the text. In addition, in renowned scientific journals, generally such curves are not reported together with ORAC values, as they are unnecessary when ORAC values are displayed.
9) English
- p.3, line 118: The sentence must begin with a subject followed by a verb.
Answer
To comply with the reviewer’s request, we have modified the sentence in line 118 (line 120 revised version) as follows:
“A suspension of DCC (450 mg, 2.18 mmol) in dry CH3CN (10 mL) was added dropwise to a mixture of IDE (500 mg, 1.48 mmol), PCA (228 mg, 1.77 mmol), and DMAP (18 mg, 0.15 mmol) in dry CH3CN (25 mL), at 0 °C”.
- p.5, line 163: compounds
Answer
This typo has been corrected.
- p.5, line 189: Photon Correlation Spectroscopy
Answer
This typo has been corrected.
Reviewer 2 Report
The manuscript describes the preparation, characterization and in vitro and in vivo tests of SLN loaded with antioxidant prodrug. Although the work is interesting and experimental section is well described, some criticisms can be raised.
1) line 28 it should be better to add keywords different from words in the title
2) in the introduction or discussion sections should be reported previous papers describing topical delivery of idebenone (eg Journal of Pharmaceutical Investigation (2013) 43:287–303). As well as previous PCA esters (eg USP 4762851 (1988)
3) line 100 DMSO-d was not reported in 2.1 materials
4) line 126 TLC elution conditions have not been reported
5) Table 1. The structures can be removed (already shown in scheme1)
6) line 165 formulations
7) line 167, about the preservative agent. Since the preparation is a leave on gel, this preservative is not allowed anymore (see Regulation 1003/2014).
8) regarding SLN preparation, a HPLC or TLC test should be required to check IDEPCA stability, because IDEPCA is a novel molecule.
9) line 180 Provide a good quality original TEM image of IDEPCA SLN. Figure 1 is an image already published (DOI: 10.3390/molecules22060887)
10) line 292. The gel amount applied in in vivo tests was very small (2 mg) and the area of treated skin was not reported. Indeed, 2 mg with 1% of active ingredient is a very low amount.The authors should provide data (from other cosmetic formulations containing IDE) supporting this choice of this amount.
11) line 292 Authors have to justify the choice of treated skin area . The back of the hand seems unusual because is an area too exposed to repeated treatments, in particular washing. Usually, the anterior region of the forearm is the preferred area or, in any case, a more "repaired" districts and even a little wider, especially when several products have to be tested at the same time.
12) line 353. Because of the novelty of IDEPCA, the stability of this molecule after 7 days of incubation should be checked and reported.
Author Response
Reviewer 2
The manuscript describes the preparation, characterization and in vitro and in vivo tests of SLN loaded with antioxidant prodrug. Although the work is interesting and experimental section is well described, some criticisms can be raised.
1) line 28 it should be better to add keywords different from words in the title
Answer
To comply with the reviewer’s request, we added the following key words: nanocarriers; anti-glycation activity; ORAC assay; skin hydration. We removed the key word “topical efficacy”.
2) in the introduction or discussion sections should be reported previous papers describing topical delivery of idebenone (eg Journal of Pharmaceutical Investigation (2013) 43:287–303). As well as previous PCA esters (eg USP 4762851 (1988)
Answer
As suggested by the reviewer, we have added the following comment in the Results and Discussion section (line 449) citing the above-mentioned reference on idebenone topical delivery:
We loaded 1% w/w IDE into SLN as this is the highest concentration used in marketed products and has already been tested in lipid nanocarriers for topical delivery of IDE [57].
In addition, we have added the following sentence to report previous patents on PCA esters (line 471):
The better hydrating effect of IDEPCA loaded SLN is supported by previous patents on PCA esters designed to improve skin hydration [58,59].
The cited references have been added to the reference list.
3) line 100 DMSO-d was not reported in 2.1 materials.
Answer
As requested, we added DMSO-d6 used for NMR analysis in the materials section.
4) line 126 TLC elution conditions have not been reported
Answer
As requested, TLC elution conditions were added (line 105) specifying the solvent used (EtOAc) as eluent.
5) Table 1. The structures can be removed (already shown in scheme1)
Answer
According to the reviewer’s suggestion, we removed the structures from Table 1.
6) line 165 formulations
Answer
This typo has been corrected.
7) line 167, about the preservative agent. Since the preparation is a leave on gel, this preservative is not allowed anymore (see Regulation 1003/2014).
Answer
We are aware that the use of the preservative Acnibio® is no longer authorized in leave-on products. However, we used the formulations containing this preservative only for research purposes. These formulations were not meant for clinical trials or for commercial purposes. If clinical trials have to be performed on these formulations, this preservative would be easily changed for water-soluble preservatives authorized under the European law.
8) regarding SLN preparation, a HPLC or TLC test should be required to check IDEPCA stability, because IDEPCA is a novel molecule.
Answer
We thank the reviewer for the suggestion. We checked IDE derivative stability by TLC as previously reported (see reference 10). Therefore, we added in the text the following sentences (line 179):
Thin layer chromatography (TLC) analyses were performed to assess IDE and its derivative stability under these conditions, as previously reported [10]. TLC analyses were carried out as described in section 2.2. chemistry. TLC analyses pointed out that no degradation of IDE or IDEPCA occurred under these conditions.
9) line 180 Provide a good quality original TEM image of IDEPCA SLN. Figure 1 is an image already published (DOI: 10.3390/molecules22060887).
Answer
To comply with the reviewer’s request, the TEM image of IDEPCA loaded SLN has been added in Fig. 1.We replaced TEM image of IDE loaded SLN with a new one. We have modified the legend of Fig. 1 accordingly.
10) line 292. The gel amount applied in in vivo tests was very small (2 mg) and the area of treated skin was not reported. Indeed, 2 mg with 1% of active ingredient is a very low amount. The authors should provide data (from other cosmetic formulations containing IDE) supporting this choice of this amount.
Answer
We applied the same amount of topical formulation previously reported in literature in in vivo studies on idebenone topical delivery (Salunkhe, S.S.; Bhatia, N.M.; Pokharkar, V.B.; Thorat, J.D.; Bhatia, M.S. Topical delivery of Idebenone using nanostructured lipid carriers: evaluations of sun-protection and anti-oxidant effects. J Pharm Investig. 2013, 43, 287-303. DOI: 10.1007/s40005-013-0079-y).
The amount applied was 2 mg/cm2. We amended the text inserting the surface area of treated skin.
11) line 292 Authors have to justify the choice of treated skin area . The back of the hand seems unusual because is an area too exposed to repeated treatments, in particular washing. Usually, the anterior region of the forearm is the preferred area or, in any case, a more "repaired" districts and even a little wider, especially when several products have to be tested at the same time.
Answer
We used the same protocol reported in previous work (see references 29, 30). The formulations investigated in this work could be useful in the treatment of skin aging. Therefore, the best choice to test their topical effect would be the application on the face. Unfortunately, it not that easy to find volunteers who accept to stop applying the products they normally use (creams, make-up etc.) to participate in the study. As the results obtained after applying the formulations under investigation on the forearm could be affected by the occlusive effect of clothes, the forearm and more “repaired” districts were discarded. On the contrary, the back of the hand, as well as the face, is constantly exposed to environmental agents and most volunteers do not use specific cosmetic hand products. Therefore, we chose the back of the hand to test the hydrating effect of the formulations under investigation.
12) line 353. Because of the novelty of IDEPCA, the stability of this molecule after 7 days of incubation should be checked and reported.
Answer
Although we did not report these experiments in the manuscript, we determined IDEPCA stability in aqueous medium during storage for 15 days at room temperature prior to prepare gel formulations. Therefore, we added in the materials and methods section (line 159) the following sentences:
“Three aliquots of IDEPCA water saturated solution were stored at room temperature for 15 days to assess IDEPCA stability in aqueous medium. At intervals (0, 1, 3, 5, 7, 15 days), samples of these solutions were withdrawn and IDEPCA content was assayed spectrophotometrically as reported above.”
We reported the results of stability tests in the results and discussion section (line 357) as follows:
“IDEPCA proved stable in water during storage at room temperature for 15 days as no significant change of IDEPCA content was observed during this period (data not shown).”
Reviewer 3 Report
The authors describe preparation of idebenone ester and its loading to lipid nanocarrier. Antioxidant as well as antiglycation activities were studied. Hydration properties of ester are much better in comparison to idebenone alone.
The manuscript is well written and experiments well designed.
Some suggestions: Pyroglutamic acid and it´s chemical name should be presented in introduction. Was IR measured as NaBr disks? Intensities of the peaks should be given (s, m, w). Conclusions part seems to be a bit short. Please, correct pytoglutamic in the last sentence.
Author Response
Reviewer 3
The authors describe preparation of idebenone ester and its loading to lipid nanocarrier. Antioxidant as well as antiglycation activities were studied. Hydration properties of ester are much better in comparison to idebenone alone.
The manuscript is well written and experiments well designed.
Some suggestions:
Pyroglutamic acid and it´s chemical name should be presented in introduction.
Answer
As suggested by the reviewer, we added the chemical name of pyroglutamic acid (2-pyrrolidone-5-carboxylic acid) in the introduction (line 52).
Was IR measured as NaBr disks?
Answer
A drop of the sample (oil) was examined directly as a thin film, "neat", between two NaCl plates. Thus, section 2.2. Chemistry has been corrected as follow:
“Infrared spectrum was recorded on a Perkin Elmer series FTIR 1600 spectrometer (Milan, Italy). A sample droplet was placed between two disks of pure NaCl (neat sample). Signal intensity were characterized as s (strong), m (medium), w (weak).
Intensities of the peaks should be given (s, m, w).
Answer
As suggested, the intensities of the peaks have been assigned:
IR (neat, selected lines) cm-1 2927 (s), 2854 (m), 1742 (s), 1704 (s), 1651 (s), 1610 (s), 1456 (m), 1266 (s), 1204 (s), 1156 (w).
Conclusions part seems to be a bit short.
Answer
According to the instructions for authors, “the conclusion section is not mandatory, but can be added to the manuscript if the discussion is unusually long or complex.” As we reported and discussed different in vitro tests and in vivo experiments, we decided to add a conclusion section. However, we summed up our results, trying to be as concise as possible and highlighting only the main results of our work.
Please, correct pytoglutamic in the last sentence.
Answer
This typo has been corrected.
Round 2
Reviewer 2 Report
Authors fully answered to the criticisms and modified the manuscript as required.
Author Response
We thank the reviewer for reviewing our manuscript.